# Recreational Green Space Service in the Guangdong–Hong Kong–Macau Greater Bay Area: A Multiple Travel Modes Perspective

Chen Weng [1,2], Jingyi Wang [1,2], Chunming Li [1,*], Rencai Dong [2,3], Chencan Lv [2,3], Yaran Jiao [1,2] and Yonglin Zhang [3]

1   Key Lab of Urban Environment and Health, Institute of Urban Environment, Chinese Academy of Sciences, Xiamen 361021, China
2   University of Chinese Academy of Sciences, Beijing 100049, China
3   State Key Laboratory of Urban and Regional Ecology, Research Center for Eco-Environmental Sciences, Chinese Academy of Sciences, Beijing 100085, China
*   Correspondence: cmli@iue.ac.cn; Tel.: +86-138-6049-1849

**Abstract:** Recreational green space (RGS) offers the most intuitive place for urban residents to get in touch with nature. The service radiation of RGS is related to the travel mode, however, residents' travel behavior has long been ignored in the study of RGS services. This paper considers the Guangdong–Hong Kong–Macau Greater Bay Area (GBA), uses multi-source data, refines the spatial distribution of residents, extracts and classifies the RGS into three categories (township (TRGS), country (CRGS), and urban (URGS)), and analyzes the spatial distribution of the three types of RGS. Using the travel isochrone, the RGS services coverage (including spatial and population coverage) in 11 cities within the GBA is defined by multiple travel modes. Finally, a comprehensive evaluation of the RGS services in the GBA is conducted based on the residents' transportation choice willingness and recreational selection. The results showed that: (1) TRGSs are mainly distributed in the suburbs, URGSs are mainly concentrated in the mid-western and southern regions, and CRGSs are mostly concentrated in the center of the GBA. (2) For daily travel (15-min and 30-min travel modes), the coverage of the RGS services is unevenly distributed, while under the 60-min travel mode, the RGS services can almost fully cover the residents in the GBA ($SP^{URGS}$ > 99%). (3) The RGS service of the central cities (Hong Kong, Macau) is better than that of the edge cities (Zhaoqing, Jiangmen), and the different city types should adopt different RGS planning and management strategies. This study provides a reference for RGS refined planning and maintenance in mega-urban agglomerations.

**Keywords:** multiple travel modes; recreational green space; green space service; Guangdong–Hong Kong–Macau Greater Bay Area

## 1. Introduction

The United Nations Population Fund (UNFPA) reports that half of the world's population now lives in urban areas, and the proportion will rise to 60% by 2030 (UNFPA, 2021). The rapid development over the past decade has made the Guangdong–Hong Kong–Macau Greater Bay Area (GBA) a region with the highest level of urbanization in China, and it joins the ranks of the four global bay areas (TBA: Tokyo Bay Area, NYBA: New York Bay Area, and SFBA: San Francisco Bay Area). The explosive urban population growth has a stress on infrastructure, which struggles to match the residents' needs, especially in developing countries [1]. Urban green spaces (UGS) are important municipal utilities, and the accessibility to green spaces helps to maintain the physical and mental health of the residents as well as improves their well-being [2]. There is no doubt that the UGS will become an increasingly important green welfare with urban development. In recent years, many mega-city clusters have prioritized UGS construction during the

new round of urban planning [3,4]. In this context, studying the service capacity of UGSs in urban agglomeration is particularly urgent and necessary from the residents' travel mode perspective.

Information about UGS is extracted from the land use and land cover datasets, which objectively reflects the spatial distribution of green spaces in urban areas [5]. However, the UGSs are not fully available for the residents' recreational activities. Within the broader category of the UGS, recreational green space (RGS) provides a place for residents' recreation activities such as nature engagement, social entertainment, and physical activity [6]. Access to an RGS in daily life is associated with better mental health, stress relief, and an overall sense of happiness [7]. Different types of RGSs are adapted for various sightseeing uses and cannot substitute for each other. In 2012, the Chinese government proposed ensuring that residents can reach green spaces within 300 m and parks within 500 m of themselves [8]. The "Outline Development Plan for the Guangdong–Hong Kong–Macau Greater Bay Area" also stated that more leisure options should be developed during urban planning to enrich the residents' leisure choices [9].

The public facility allocation based on residential travel behaviors has been proven to be a mature way to improve the life quality of the residents since it meets the residents' needs [10]. Scholars divide urban traffic circles into three categories according to temporal scales: quarter-hour, half-hour, and one-hour types [11,12]. Different traffic circles correspond to different travel modes at different spatial scales. The spatial analysis in traffic circles studies is usually based on the buffer area of uniform expansion, and it ignores the influence of the actual road distribution. In reality, the boundaries of the traffic circles depends on the road network, and they varies significantly among regions. A traffic circle can be represented by a travel isochrone (TIC), an irregular polygon that is surrounded by points of equal travel time around a given location [13]. The TIC depends on road networks that can objectively define the geographic range of the traffic circles and be used to delineate the travel range of the residents, accurately [13].

Currently, most of the studies of the relationship between the residents' UGSs have been based on accessibility approaches with the help of the Geographic Information System (GIS). For instance, the minimum cumulative resistance model and origin–destination matrix estimation model have been widely used to analyze the spatial patterns and evaluate the UGS' suitability [14,15]. Their data granularity is, however, singular and ignores the dynamic variability of green space accessibility via different travel modes. The two-step floating catchment area method (2SFCA) has been developed in recent years to analyze the relationship between the urban residents and the elements, and it can be used to assess the public facilities under different spatiotemporal conditions [16]. The 2SFCA method considers the UGS as a supplier and residents as demanders; it allows for the analysis of the accessibility between them through a radius search, ignoring the differences in the accessible radii in different regions. Online map services that possess a high resolution, accurate time estimates, and the integration of multiple traffic modes, such as Google Map, Baidu Map, and Mapbox, provide the possibility for a customized analysis to be performed of the spatial relationship between the urban elements.

The residents' travel behavior should be considered in urban green space planning, which has been ignored in current studies. This study chose the GBA as the study area to evaluate the RGS services from a multiple travel mode perspective: the extraction and classification of green spaces with recreational value (RGS); the analysis of RGS service coverage through the isochronous circle (TIC) method; the RGS service evaluation under multiple travel modes. The study result can provide help for RGS refined planning and maintenance in mega-urban agglomerations.

## 2. Materials and Methods

### 2.1. Study Area

With its rapid development in recent years, the GBA has become one of the world's four major bay areas, with it joining the NYBA, TBA, and SFBA. The GBA created more

than 1.96 trillion U.S. dollars of gross domestic product (GDP) in 2021 (contributing about 1/10 of China's national GDP), and its resident population reached about 86 million. In 2019, the Chinese government released the Outline Development Plan for the Guangdong–Hong Kong–Macau Greater Bay Area, promoting the construction of the transportation network in the GBA. By the end of 2020, the total length of the expressway in the GBA exceeded 5000 km [17], and the road network density reached 0.082 km/km$^2$ (it was ranked first among the four major bay areas). The completion of the "one-hour traffic circle" has dramatically improved the travel efficiency for the residents in the GBA [18]. Meanwhile, as one of the world's most economically developed and densely populated areas, the GBA also has one of the highest levels of conflict between the people and the urban green space. Therefore, this study selects the GBA as an example to find the characteristics of the RGS service capacity in bay area urban agglomerations and provide a reference for RGS planning and the improvements that can be made to the residents' quality of life.

The RGS service capacity is related to the residents' travel behavior. This study analyzed the service capacity of the RGSs within different traffic circles and travel modes. Urban, county, and township are selected as the spatial division units for the research scales in this study according to the division of the traffic circle scale [12]. These three units represent the administrative hierarchy's primary organizational levels, which are controlled in a top–down manner in the Chinese system of urban government. They also correspond to the micro, meso, and macro aspects of the study of urban problems. A total of nine cities (Dongguan, Foshan, Guangzhou, Huizhou, Jiangmen, Shenzhen, Zhaoqing, Zhongshan, and Zhuhai), two special administrative regions (Hong Kong and Macau), seventy-six counties, and six hundred and thirty townships were included in this study (Figure 1).

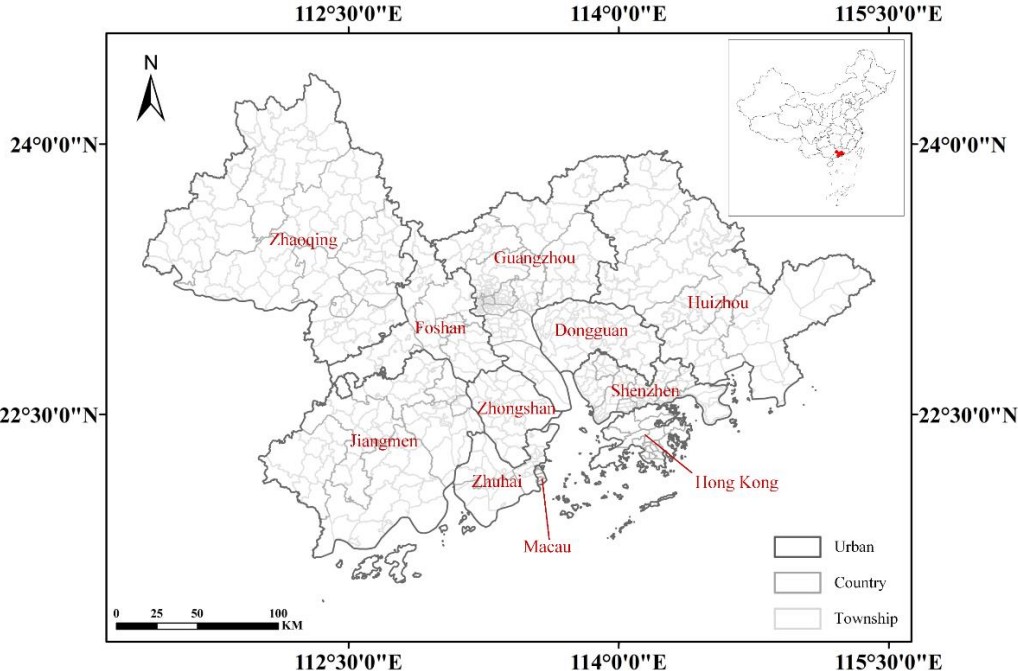

**Figure 1.** Study area and its location in China.

## 2.2. Data and Methods

Figure 2 presents the RGS service evaluation workflow, which is divided into five parts. Firstly, research data were obtained through multi-source data collection and processing methods. Secondly, the population distribution was spatialized to residential points based on the building contours. Thirdly, the recreational green space (RGS) was extracted and classified into three categories (a township-type RGS, a country-type RGS, and an urban-type RGS), and their spatial distribution was analyzed. Fourthly, the online mapping

service was used to delineate the travel isochronous (TIC), and the spatial statistics analysis method was used to calculate the coverage of RGS service (area ratio and population ratio). Then, the RGS service population coverage per unit area was analyzed by a spatial join analysis. Finally, the RGS service was comprehensively evaluated based on the residents' transportation choosing willingness and recreation choice probability.

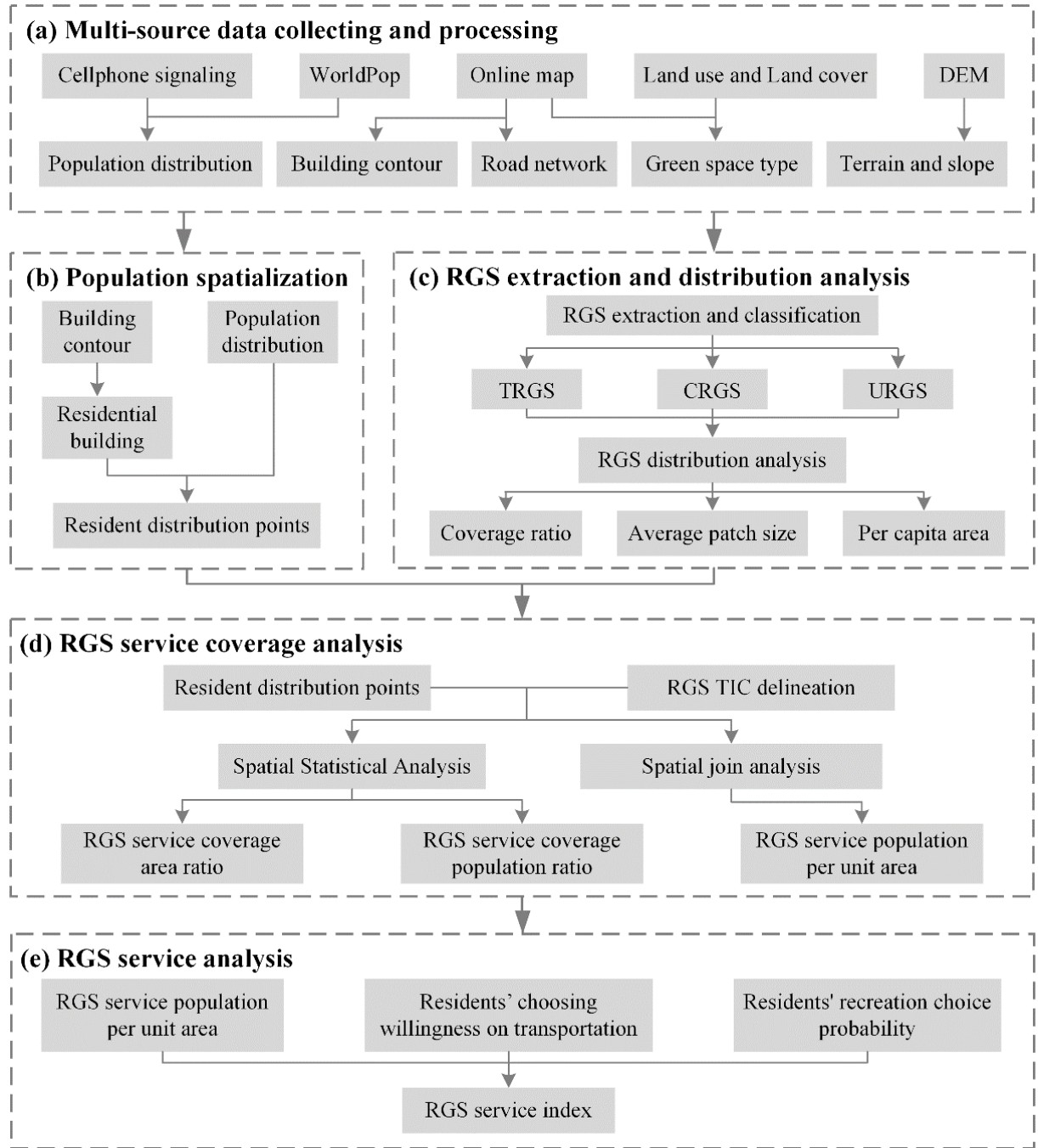

**Figure 2.** RGS service evaluation in the GBA. Note: DEM, Digital Elevation Model; RGS, recreational green space; TRGS, township-type recreational green space; CRGS, country-type recreational green space; URGS, urban-type recreational green space; TIC, travel isochrone.

### 2.2.1. Population Spatialization

The spatial distribution pattern of the population in the GBA was estimated using multi-source data (Figure 3). This study selected the 250 m resolution population dis-

tribution raster data from China Unicom (obtained from the expanded sample statistics of mobile phone signaling) and the 100 m resolution population distribution raster data from WorldPop to estimate the population distribution. The data were selected from October to November 2019 (the spatial population distribution was not affected by the COVID-19 outbreak), as China has gradually eased the travel restrictions and returned to the pre-pandemic state.

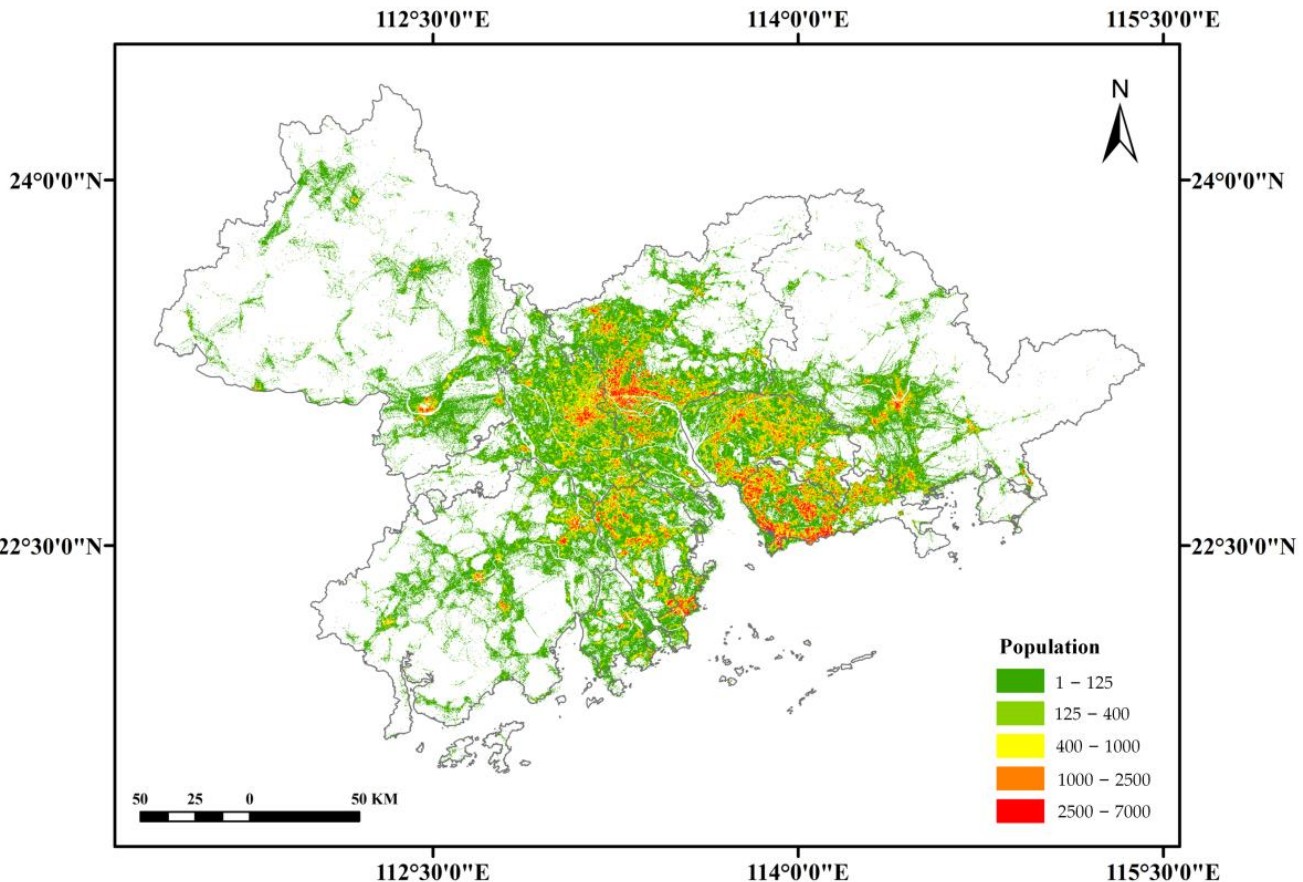

**Figure 3.** Population distribution in the GBA.

By referring to the peanutButter web application in WorldPop [19], this study constructed a population distribution refinement method to refine the population to the exact geographical coordinate points (Figure 4). Building contours were extracted from online maps, including Baidu Maps, AMAP, Map World, and Open Street Map. The "Join Attributes by Location tool" in QGIS (V3.20.3) was used to process all of the building contour data layers and determine the overlapping properties of the layers (Figure 4a). These layers were then merged, and the redundant building polygons were removed by filtering the contained relations and statistics in the overlapping properties of each polygon (Figure 4b). The non-residential building polygons were then removed using the "Type" field in the layer properties (Figure 4c). Finally, in Figure 4d, the centroid of the residential buildings was extracted, and the population raster data (a darker red color means more people) were evenly disaggregated among the residential building points (for cells without residential building points, the centroid was taken as the additional point). The population distribution raster data from 348,817 effective cells were refined into 3,302,974 resident distribution points using this method.

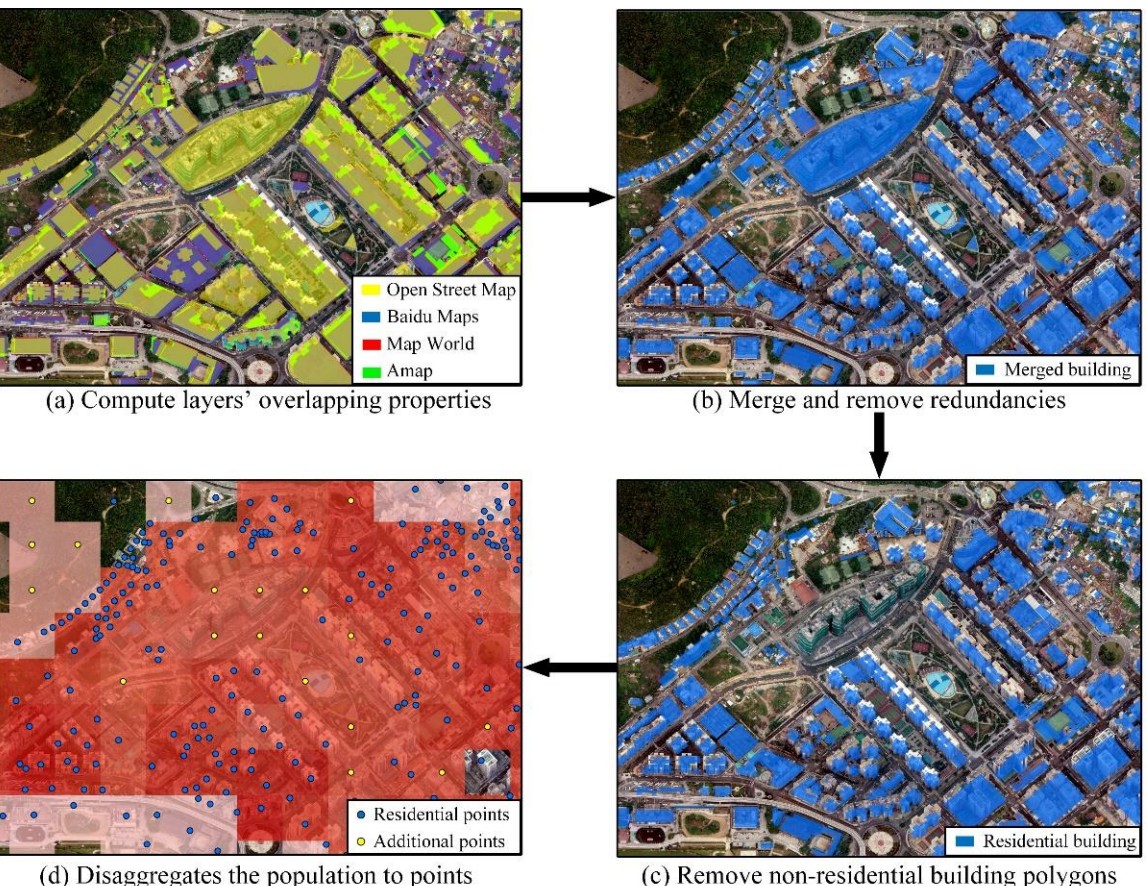

**Figure 4.** The process of allocating the population to residential points.

2.2.2. RGS Extraction and Distribution Analysis

Extraction and Classification

Green spaces provide an important venue for the residents' recreation activities, such as exercise, leisure, entertainment, culture, and other services. This study adopted the multi-source data fusion method (including data related to land use, land cover, online maps, digital elevation models (DEMs), and road networks) to obtain the RGS data for the GBA (Figure 5). The land use data came from China's third national land survey, and it covers 12 broad categories and 73 different land-use types. The land cover data were obtained from the GlobalLand30 dataset of the Chinese National Catalogue Service for Geographic Information. The road network and green space data were extracted from current mainstream online maps using online mapping platforms (Baidu Maps, AMAP, Map World, and Open Street Map). The DEM data came from the Geographical Information Monitoring Cloud Platform (http://www.dsac.cn/ (accessed on 20 September 2020)). Different types of green spaces provide the different types of recreation services [20], and the residents have different recreational demands under different travel modes [18]. Referring to the residents' recreational options, this study divided the RGS into three categories (Table 1): the urban-type RGS (URGS), the country-type RGS (CRGS), and the township-type RGS (TRGS) [21]. The URGSs are those with high requirements for visiting time, such as national parks, forest parks, scenic spots, etc., which primarily provide services for the residents to conduct occasional short trips during holidays, and they usually have the highest landscape or visitation value [22,23]. The CRGSs include urban parks, green squares, green sports fields, etc., which meet the residents' needs for sports, leisure, and social interaction in their daily leisure time; they are an essential element of municipal public facilities, and each of them has an area of more than 1000 square meters and a continuous and complete spatial morphology [24]. The TRGSs are those small



or undeveloped green spaces, such as parterres, pocket parks, center street parks, etc., which do not require much space and are more flexible in urban planning and construction designs [6,25]. Although they have a low visitation value, TRGSs are widely distributed and easily accessible, making them a good place for residents to take a short break [26].

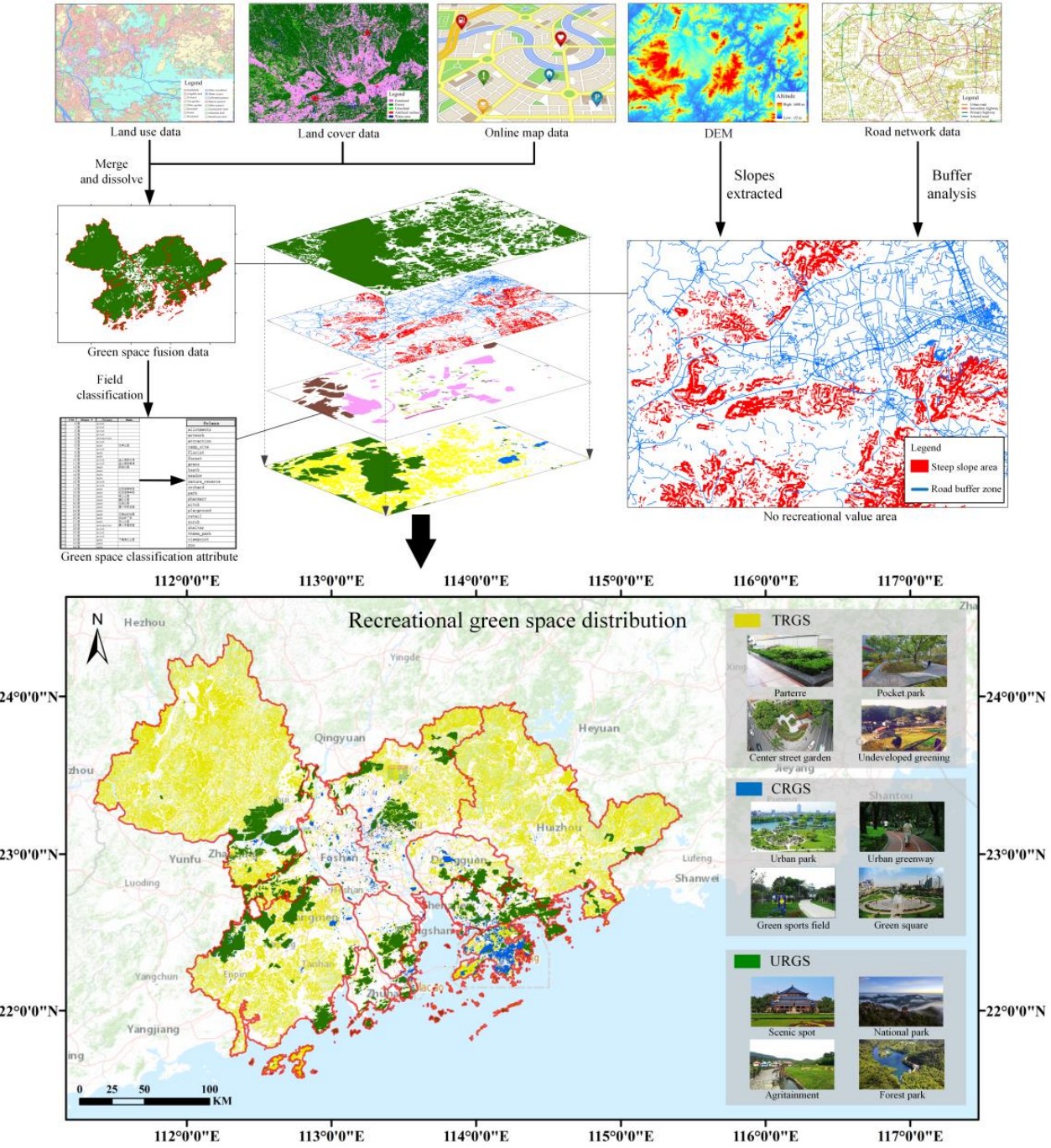

**Figure 5.** Recreational green space extraction and classification.

**Table 1.** RGS statistics.

| Category | Service Type | Green Space Types | Total Area (km$^2$) |
|---|---|---|---|
| TRGS | Brief respite | Pocket park, Center street park, Parterre, Undeveloped greening | 142,252.75 |
| CRGS | Daily leisure | Urban park, Urban greenway, Green sports field, Green square | 16,059.62 |
| URGS | Leisure tour | Scenic spot, National park, Agritainment, Forest park | 79,544.17 |

Note: RGS, recreational green space; TRGS, township RGS; CRGS, country RGS; URGS, urban RGS.

As shown in Figure 5, the data were processed as follows: Firstly, QGIS (V3.20.3) was used to merge and dissolve the green space data from different sources and classify the data fields into three types according to the service types. Secondly, the areas that were unsuitable for hiking (slope > 25%) were eliminated based on DEM by referring to the study of mountain exercise physiology [27]. Thirdly, the buffer zone of the road network was extracted to remove the road greenbelts according to The Classification Standard of Urban Green Space [28]. Finally, non-RGSs were removed from the green space data, and the RGS distribution was obtained in the GBA. Among the three categories of green spaces, the proportion of CRGSs is the smallest, accounting for only 6.75% of the total amount, while the proportions of URGSs and TRGSs were 33.44% and 59.81%, respectively (Table 1).

Distribution Analysis

This study identified the coverage ratio ($CR$), average patch size ($APS$), and per capita area ($PCA$) as indicators to explore the distribution pattern of RGS in the GBA. The equations of these three indicators are shown as follows:

$$CR_h = \frac{\sum_{i=1}^{n} S_i^k}{S_h} * 100\% \tag{1}$$

where $CR_h$ represents the RGS coverage ratio in the administrative division unit $h$; $k$ denotes the category of RGS that was analyzed (TRGS, CRGS, or URGS); $S_h$ is the area of the administrative division unit $h$; $n$ denotes the number of RGS patches in the administrative division unit $h$; $S_i^k$ is the area of the $i$-th RGS patch.

$$APS_h = \frac{\sum_{i=1}^{n} S_i^k}{n} \tag{2}$$

where $APS_h$ represents the average area of the RGS blocks in the administrative division unit $h$, which reflects the fragmentation and landscape heterogeneity of the RGS [29]. A higher value means more concentrated green space, and a lower value means more dispersed green space.

$$PCA_h = \frac{\sum_{i=1}^{n} S_i^k}{P_h} \tag{3}$$

where $PCA_h$ is the per capita RGS area in the administrative division unit $h$, which is an indicator to measure the matching degree between the RGS and the population, and $P_h$ is the population in the administrative division unit $h$.

2.2.3. RGS Service Coverage Analysis

The RGS service is related to the residents' travel modes and needs to be analyzed based on a travel isochrone (TIC). Firstly, we delineated the TIC based on the travel modes of residents from the centroids of the RGS patches. Then, the TIC of the RGS was overlaid with the population distribution and the administrative divisions, and the space and population proportions covered by the TIC of the RGS were then calculated. Finally, spatial

join analysis was used to calculate the population that could be served per unit area of each RGS patch.

Isochrone Delineation

A TIC is a geospatial reflection of the traffic time, and it is often used to quantify the range of accessibility without defining a specific purpose [30]. Referring to prior studies on traffic circles and green space accessibility [31], this study matched the corresponding travel modes to three kinds of RGS (TRGS, CRGS, and URGS) and assigned them to six TICs (Table 2), as illustrated in Figure 6. This study used the centroid of the RGS patch as the starting point to delineate the isochrone. Large RGS patches were split into multiple smaller ones (at intervals of a 1-min travel range under selected travel modes) to improve the accuracy of the TIC delineation. Finally, the Mapbox online map service API (https://docs.mapbox.com/api/navigation/isochrone/ (accessed on 8 December 2021)) was used to obtain the TIC data for all of the RGS blocks in the GBA.

**Table 2.** Average TIC attribute values.

| Travel Mode | RGS Category | TIC Radius (km) | TIC Area (km²) |
|---|---|---|---|
| 15 min walking | TRGS | 0.47 | 0.70 |
| 15 min cycling | TRGS | 1.17 | 4.31 |
| 30 min walking | CRGS | 1.37 | 5.90 |
| 30 min cycling | CRGS | 4.27 | 57.32 |
| 30 min driving | CRGS | 14.39 | 650.64 |
| 60 min driving | URGS | 39.38 | 4870.964 |

Note: TIC, travel isochrone; RGS, recreational green space; TRGS, township RGS; CRGS, country RGS; URGS, urban RGS.

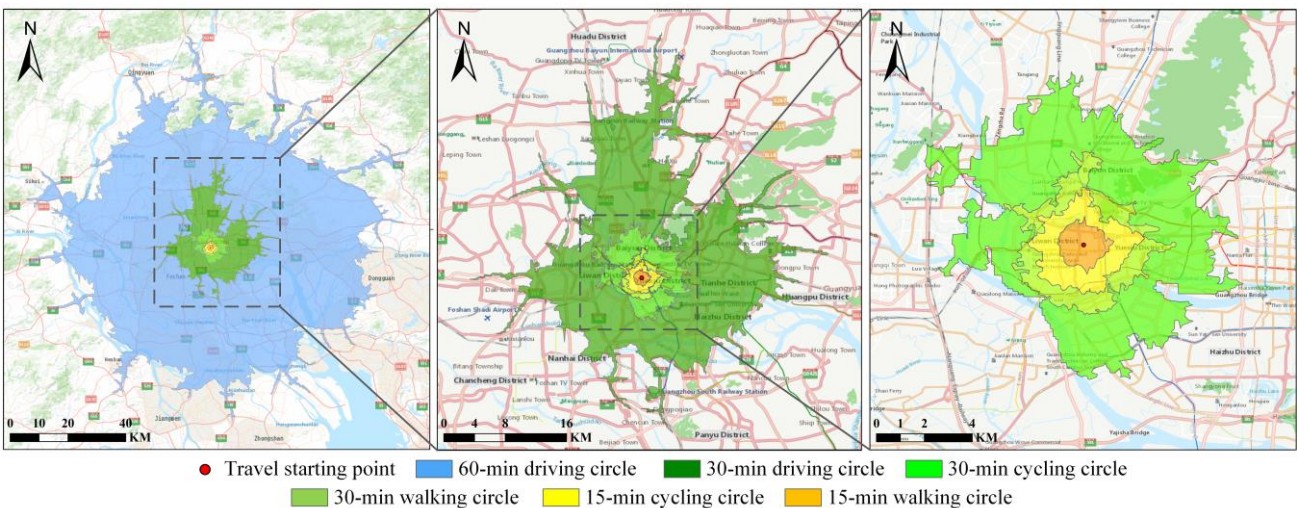

**Figure 6.** TIC delineation of different travel modes.

Coverage Analysis

Space syntax was used to discover the implicit relationships between the features through the topological relationships [32], and it is widely used to study the relationship between human beings and the landscape [33]. This study constructed a method for studying the spatial relationship between the RGS service and the residents based on the space syntax. The administrative divisions of the urban, country, and township areas were used as computing units to calculate the service space coverage ratio (Equation (4)) and the population coverage ratio (Equation (5)) of different categories of RGSs.

$$SC_h^k = \frac{\sum_{i=1}^{n}\left(S_i^{TIC} - S_i^{oTIC}\right) + \sum_{i=1}^{m} S_i^{IN}}{S_h} * 100\% \tag{4}$$

where $h$ is the administrative division unit; $SC_h^k$ is the service space coverage of the $k$-type RGS in the computed administrative division unit $h$; $n$ is the number of RGS patches in $h$; $S_i^{TIC}$ is the area of the TIC in the RGS patch $i$; $S_i^{oTIC}$ is the overlapping area of the TICs in the RGS blocks $i$; $m$ is the number of intersecting TIC blocks in $h$; $S_i^{IN}$ is the area of the intersecting patch $i$ of the TIC; $S_h$ is the area of $h$.

$$SP_h^k = \frac{\sum_{i=1}^{n} P_i}{P_h} * 100\% \tag{5}$$

where $SP_h^k$ is the RGS service population coverage of the $k$-type RGS in the administrative division unit $h$. A higher value means more people can enjoy the RGS services; $n$ is the number of resident distribution points covered by the RGS's TIC in $h$; $P_i$ is the population of the $i$-th resident distribution point; $P_h$ is the total population in the administrative division unit $h$.

A spatial join analysis was used to calculate the number of residents covered by the service per square meter of RGS ($PCRS$) by overlaying its TIC with the residential points. The calculation of $PCRS$ is shown in Equation (6):

$$PCRS_g^k = \frac{\sum_{i=1}^{n} P_i}{S_g} \tag{6}$$

where $PCRS_g^k$ denotes the number of residents covered by the $k$-type RGS patch $g$; $n$ denotes the number of residential points covered by the TIC of the RGS patch $g$ $P_i$ represents the population of the $i$-th TIC covered the residential point; $S_g$ is the area of the RGS patch $g$.

### 2.2.4. Service Analysis

This study used a location entropy calculation to analyze the level of the RGS services of each township in the GBA. The services of each category of RGS were then evaluated based on the residents' transportation choice willingness. Finally, the RGS services of the township were evaluated comprehensively according to the residents' RGS selection probability.

Location Entropy Calculation

Geospatially, location entropy reflects the level of a spatial unit in the entire domain, and it is widely used to analyze the equilibrium of the spatial data [34]. This study used a location entropy calculation to analyze the RGS service level and the equilibrium of the township units in the GBA. The calculation formula is shown in Equation (7).

$$ERS_h^k = \frac{\sum_{i=1}^{n} P_i}{PCRS_{GBA}^k \sum_{i=1}^{n} S_i} \tag{7}$$

where $ERS_h^k$ denotes the service level of the $k$-type RGS in the township unit $h$. If $ERS_g^k > 1$, this township unit has a high RGS service capacity. If $ERS_g^k < 1$, the RGS service capacity of this township unit is considered to be weak. The absence of an $ERS_g^k$ value indicates that this type of RGS is missing in this township unit; $n$ is the number of RGS patches in the computed township unit $h$; $P_i$ is the number of residents covered by the TIC of $i$-th RGS patch; $S_i$ is the area of the $i$-th RGS patch; $PCRS_{GBA}^k$ is the average population for each square meter of the $k$-type that the RGS can serve in the GBA.

Evaluation of RGS Service under Multiple Travel Modes

To comprehensively evaluate the service of the RGS under multiple travel modes, based on Peng et al. [35] and Yang et al. [36], this study adopted a multi-factor evaluation model and constructed an RGS service index (Equation (8)). Finally, the three categories

of RGS were integrated, and the comprehensive service index of the RGS service was calculated according to the residents' recreation choice probability (Equation (9)) [12].

$$RSE_h^k = \frac{\sum_{i=1}^n \left( ERS_i^k * W_i \right)}{\sum_{i=1}^n W_i} \tag{8}$$

where $RSE_h^k$ denotes the $k$-type RGS's service evaluation index at multiple travel modes in the computed township unit $h$; $n$ is the number of travel modes that are involved in the $k$-type RGS service; $ERS_i^k$ is the service level of the $k$-type RGS under the $i$-th travel mode; $W_i$ is the proportion of the $i$-th travel mode in the RGS service. The weight of transportation in this study ($W_{walking} = 0.67$; $W_{cycling} = 0.15$; $W_{driving} = 0.18$) is based on the residents' willingness to choose a certain transportation mode [23].

$$RSE_h^{RGS} = 0.6 * RSE_h^{TRGS} + 0.35 * RSE_h^{CRGS} + 0.05 * RSE_h^{URGS} \tag{9}$$

where $RSE_h^{RGS}$ denotes the comprehensive service evaluation index of three categories RGS in the computed township unit $h$; $RSE_h^{TRGS}$, $RSE_h^{CRGS}$, and $RSE_h^{URGS}$ represent the service evaluation index of TRGS, CRGS, and URGS, respectively.

Finally, the RGS service evaluation index was divided into five grades (high, relatively high, medium, relatively low, and low) by a quantile method, and this was visualized by hierarchical coloring.

## 3. Results

### 3.1. RGS Distribution

The total coverage of the RGS is 53.46% in the GBA. The distribution of the RGS in the GBA is uneven, and the area of three categories of the RGSs vary greatly (CR$_{TRGS}$ = 32.50%; CR$_{CRGS}$ = 3.67%; CR$_{URGS}$ = 18.17%). In general, the allocation of TRGS in the GBA is negatively correlated to the urbanization development level (Figure 7) due to the negative impact of urban development on the vegetation [37]. Hong Kong (CR$_{TRGS}$ = 55.7%; CR$_{CRGS}$ = 5.3%) and Macau (CR$_{TRGS}$ = 20.6%; CR$_{CRGS}$ = 19.0%) have a large proportion of TRGSs and CRGSs, but they are poor in terms of the RGS area per capita (Hong Kong PCA$_{TRGS}$ = 748.9 m$^2$/p, PCA$_{CRGS}$ = 225.7 m$^2$/p; Macau PCA$_{TRGS}$ = 129.6 m$^2$/p, PCA$_{CRGS}$ = 33.4 m$^2$/p). The average RGS patch size is also small (Hong Kong APS$_{TRGS}$ = 0.6 km$^2$, PCA$_{CRGS}$ = 1.3 km$^2$; Macau PCA$_{TRGS}$ = 0.1 km$^2$, PCA$_{CRGS}$ = 0.2 km$^2$), and this is perhaps caused by long-term urbanization in these two cities. These two cities have a long history of construction, and the developed urban built-up areas have a high building density, therefore, the RGS can only be built in small open spaces. Zhaoqing (CR$_{TRGS}$ = 55.7%) and Huizhou (CR$_{TRGS}$ = 41.4%) have a high proportion of TRGS coverage because these two cities are located in the mountainous area on the edge of the GBA, and their undeveloped green space is abundant.

Figure 8 shows the distribution of three categories of RGSs in the GBA; the TRGS are widely distributed in the GBA. They are mainly at the region's margins, while CRGSs and URGSs are mainly spread around the built-up area of the urban agglomeration. The CRGS *CR* is closely related to urbanization (Figure 7). When they are compared with less developed cities (Huizhou, Zhaoqing, and Zhongshan), the developed cities (Hong Kong, Shenzhen, and Macau) devote more resources to constructing urban parks and squares (CRGS). The URGSs are mainly distributed in the mid-western and southern regions in the GBA (Figure 8), and the *PCA* is inversely proportional to the population distribution.

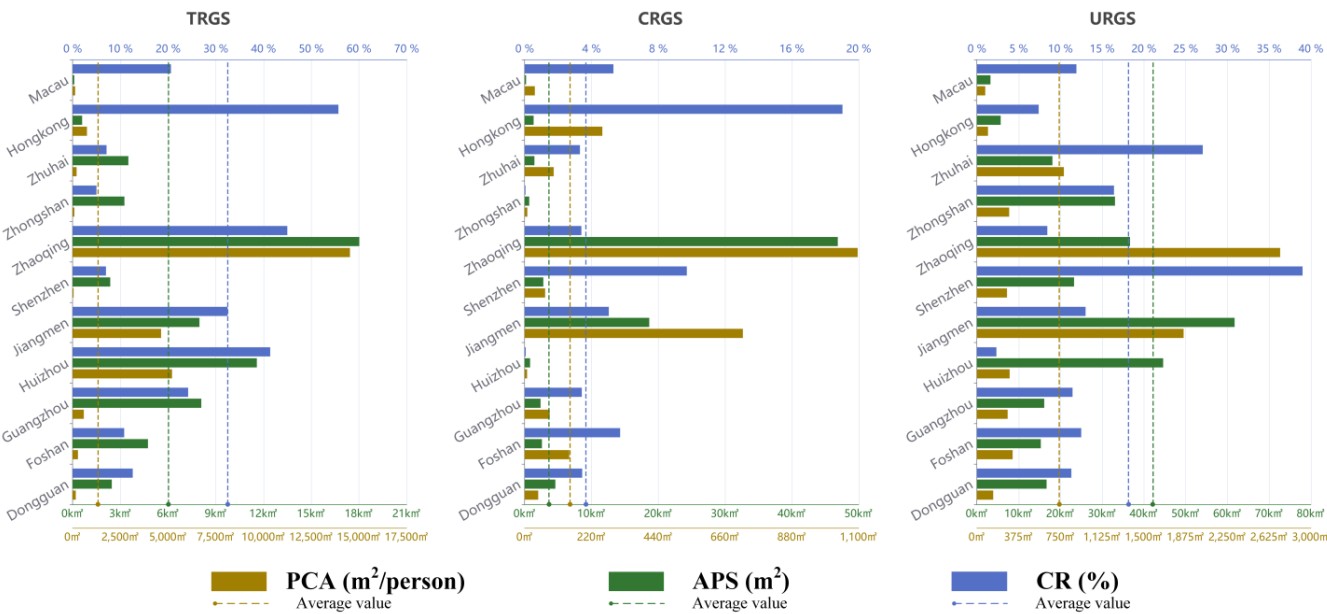

**Figure 7.** RGS statistics of 11 cities in the GBA.

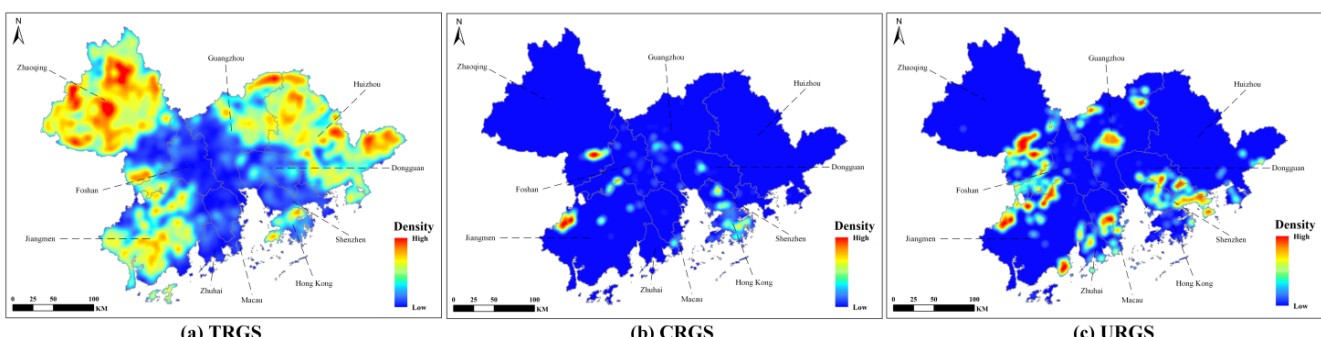

**Figure 8.** The distribution density of three categories of RGS.

*3.2. RGS Service Coverage*

In this study, we conducted spatial statistics on the space and population that are covered by the travel isochrone (TIC) of the RGS under different travel modes. For a better comparison to be made, this study uses the RGS service coverage of the whole GBA as the baseline array. In general, the service coverage of the RGS (*SC*) in the GBA is higher than the coverage of the RGS (*CR*), and the rate of improvement gradually increases as travel modes are upgraded (Table 3).

For TRGS, the overall service coverage improvement of the GBA under the 15-min walking mode reached 17.58% and it increased to 26.31% in the 15-min cycling mode. Except for Huizhou (15 min walking: 11.71%; 15 min cycling: 15.39%), Zhaoqing (15 min walking: 8.87%; 15 min cycling: 9.10%), and Jiangmen (15 min walking: 16.83%; 15 min cycling: 23.33%), the RGS service coverage improvement in the other cities is higher than that of the whole GBA in the 15-min walking mode. The main reason for this is that these three cities are located in the mountains on the edge of the GBA, and the green space resources are abundant, but concentrated (Figure 7). It is worth noting that Hong Kong has an above-baseline improvement in the 15-min walking mode (22.95%) and a below-baseline improvement in the 15-min cycling mode (23.30%). Due to the high coverage and even distribution of the TRGSs in Hong Kong, the coverage of the TRGSs in the 15-min walking mode is close to saturation (78.65%), and the 15-min cycling mode cannot identify a further improvement on this. For the CRGSs, except for Huizhou (30 min walking: 0.99%; 30 min cycling: 4.19%; 30 min driving: 27.06%), Jiangmen (30 min walking: 1.87%; 30 min cycling:

9.25%; 30 min driving: 42.77%), Zhaoqing (30 min walking: 0.75%; 30 min cycling: 1.96%; 30 min driving: 15.68%), and Hong Kong (30 min walking: 0.36%; 30 min cycling: 2.24%; 30 min driving: 24.17%), which are below the baseline (30 min walking: 4.97%; 30 min cycling: 16.72%; 30 min driving: 44.01%), the other cities in the GBA have a higher-than-baseline service coverage improvement. Huizhou, Jiangmen, and Zhaoqing have fewer CRGSs, and these are mainly concentrated in the built-up areas, and it is difficult for the services to cover the suburbs as Hong Kong's CRGS are mainly concentrated on Hong Kong Island and the Kowloon Peninsula, while the New Territories (the largest area) still have a lot of undeveloped lands, where the road network is less developed. In the 60-min driving mode, the service coverage of the URGSs in the GBA is increased by 61.44%. Only Zhaoqing is below baseline in this travel mode because it is located northwest of the GBA, with a large area and low road network density.

**Table 3.** RGS service space coverage (*CR* ■/*SC* ■).

| RGS category | TRGS | | | CRGS | | | URGS |
|---|---|---|---|---|---|---|---|
| Travel pattern | 15-min walking | 15-min cycling | 30-min walking | 30-min cycling | 30-min driving | 60-min driving |
| Dongguan | 12.59% | 12.59% | 3.44% | 3.44% | 3.44% | 11.31% |
| | 45.55% | 69.56% | 16.56% | 54.35% | 93.53% | 99.72% |
| Foshan | 10.80% | 10.80% | 5.72% | 5.72% | 5.72% | 12.50% |
| | 36.82% | 63.58% | 24.23% | 63.18% | 93.64% | 100.00% |
| Guangzhou | 24.20% | 24.20% | 3.42% | 3.42% | 3.42% | 11.45% |
| | 50.09% | 65.85% | 14.58% | 35.92% | 74.91% | 96.10% |
| Huizhou | 41.40% | 41.40% | 0.06% | 0.06% | 0.06% | 2.34% |
| | 53.11% | 56.79% | 1.05% | 4.25% | 27.12% | 78.26% |
| Jiangmen | 32.48% | 32.48% | 5.04% | 5.04% | 5.04% | 13.02% |
| | 49.31% | 55.81% | 6.91% | 14.56% | 47.81% | 91.11% |
| Shenzhen | 6.98% | 6.98% | 9.71% | 9.71% | 9.71% | 39.00% |
| | 49.34% | 65.95% | 35.29% | 64.00% | 90.02% | 99.93% |
| Zhaoqing | 44.99% | 44.99% | 1.15% | 1.15% | 1.15% | 8.44% |
| | 53.77% | 54.09% | 1.90% | 3.11% | 16.83% | 50.73% |
| Zhongshan | 4.99% | 4.99% | 0.05% | 0.05% | 0.05% | 16.42% |
| | 26.92% | 50.81% | 11.95% | 38.04% | 94.85% | 99.57% |
| Zhuhai | 7.10% | 7.10% | 3.31% | 3.31% | 3.31% | 27.05% |
| | 38.29% | 59.40% | 16.81% | 46.24% | 85.52% | 98.42% |
| Hongkong | 55.70% | 55.70% | 19.03% | 19.03% | 19.03% | 7.40% |
| | 78.65% | 79.00% | 19.39% | 21.47% | 43.29% | 76.00% |
| Macau | 20.60% | 20.60% | 5.31% | 5.31% | 5.31% | 11.92% |
| | 52.48% | 68.53% | 14.54% | 63.76% | 91.03% | 96.30% |
| GBA | 32.50% | 32.50% | 3.67% | 3.67% | 3.67% | 18.17% |
| | 50.08% | 58.81% | 8.64% | 20.39% | 47.68% | 79.58% |

Note: RGS, recreational green space; *CR*, RGS coverage; *SC*, RGS service space coverage; TRGS, township RGS; CRGS, country RGS; URGS, urban RGS.

Table 4 shows the RGS service population coverage under different travel modes for 11 cities in the GBA. In general, as the travel modes are upgraded (expansion of the TIC), the RGS service coverage gradually increases. In the 15-min walking mode, the TRGSs, which are most closely related to the people's daily life, can serve more than half of the population in the GBA ($SP^{TRGS}$ = 63.8%). Except for Zhaoqing ($SP^{TRGS}$ = 44.42%) and Zhongshan ($SP^{TRGS}$ = 45.06%), the TRGS service population coverage of other cities is higher than 50%. In the 15-min riding mode, the TRGS services in each city can cover more than two-thirds of residents. For the CRGSs, their service is uneven across cities in the 30-min walk travel mode. For Dongguan, Huizhou, Zhaoqing, and Zhongshan, the CRGS service population coverage rate is less than 40%, while in Hong Kong and Macau, it is close

to 100%. Using a 30-min travel mode can greatly reduce the difference between the cities, which indicates that upgrading the travel mode can make up for the uneven distribution of the CRGS service coverage. In the 60-min driving travel mode, the URGS services can cover almost all of the residents in the GBA ($SP^{URGS}$ = 99.31%), which means nearly all of the GBA residents can reach suitable green recreation places within an hour's drive. The results were supported by the achievements made by the government's "one-hour traffic circle" construction, which started in 2013 [9].

**Table 4.** Ratio of RGS service population coverage in each city (SP ▢).

| RGS category | TRGS | | CRGS | | URGS | |
|---|---|---|---|---|---|---|
| Travel pattern | 15–min walking | 15–min cycling | 30–min walking | 30–min cycling | 30–min driving | 60–min driving |
| Dongguan | 54.25% | 91.55% | 29.81% | 81.34% | 99.43% | 99.99% |
| Foshan | 54.85% | 92.26% | 61.09% | 94.12% | 99.97% | 100.00% |
| Guangzhou | 65.77% | 93.53% | 67.02% | 92.91% | 99.77% | 99.99% |
| Huizhou | 55.49% | 77.92% | 17.42% | 43.02% | 87.28% | 99.85% |
| Jiangmen | 64.38% | 86.19% | 46.50% | 70.90% | 95.61% | 99.93% |
| Shenzhen | 70.46% | 93.96% | 65.04% | 95.61% | 99.96% | 99.99% |
| Zhaoqing | 44.42% | 66.51% | 23.48% | 37.28% | 74.92% | 95.20% |
| Zhongshan | 45.06% | 75.00% | 35.27% | 72.16% | 99.73% | 99.98% |
| Zhuhai | 83.62% | 94.75% | 68.43% | 91.62% | 99.62% | 99.99% |
| Hongkong | 99.25% | 99.48% | 99.34% | 99.48% | 99.77% | 99.49% |
| Macau | 98.31% | 98.65% | 98.30% | 98.65% | 99.90% | 99.90% |
| GBA | 63.85% | 89.63% | 55.02% | 84.25% | 97.42% | 99.31% |

Note: RGS, recreational green space; TRGS, township RGS; CRGS, country RGS; URGS, urban RGS.

This study uses a spatial join analysis to calculate the population that can be covered per square meter of RGS, and the statistical results are shown in Table 5. Generally speaking, the RGSs in the densely populated cities (Macau, Hong Kong, and Zhongshan) are more crowded, while those in mountainous cities (Jiangmen and Zhaoqing) are deserted. The *PCRS* of all of the RGS categories in Macau is high due to its small urban area and its even population distribution. The *PCRS* of Zhaoqing and Jiangmen is low because they are located in the western mountainous areas of the GBA with inconvenient transportation links and relatively backward urbanization: the service capacity of most of the green spaces here cannot be fully utilized as they remain undeveloped.

**Table 5.** Number of residents covered by RGS service per square meter (*PCRS*).

| | TRGS | | CRGS | | URGS | | |
|---|---|---|---|---|---|---|---|
| Dongguan | 0.0377 | 0.3502 | 0.0534 | 0.4368 | 2.6026 | 7.8066 | High |
| Foshan | 0.0081 | 0.0590 | 0.0650 | 0.4798 | 3.4865 | 3.8717 | |
| Guangzhou | 0.0065 | 0.0412 | 0.0956 | 0.6774 | 4.8926 | 7.5045 | |
| Huizhou | 0.0018 | 0.0107 | 0.3823 | 1.9301 | 7.0116 | 0.6970 | |
| Jiangmen | 0.0126 | 0.0319 | 0.0077 | 0.0354 | 0.1633 | 0.2634 | |
| Shenzhen | 0.1157 | 0.7344 | 0.2148 | 1.4412 | 8.2381 | 2.9481 | |
| Zhaoqing | 0.0003 | 0.0016 | 0.0080 | 0.0270 | 0.1155 | 0.6824 | |
| Zhongshan | 0.0366 | 0.2419 | 0.8695 | 6.1458 | 22.4300 | 2.7224 | Medium |
| Zhuhai | 0.0325 | 0.1504 | 0.2210 | 1.3382 | 3.7603 | 1.4866 | |
| Hongkong | 0.8101 | 3.4179 | 0.7988 | 2.4158 | 21.8768 | 22.5815 | |
| Macau | 6.7192 | 25.6471 | 7.7366 | 18.6775 | 67.2327 | 10.7912 | |
| GBA | 0.0269 | 0.1178 | 0.1941 | 0.9227 | 6.4403 | 4.0069 | Low |
| | 15-min walking | 15-min cycling | 30-min walking | 30-min cycling | 30-min driving | 60-min driving | |

Note: GBA, Guangdong-Hong Kong-Macau Greater Bay Area; RGS, recreational green space; TRGS, township RGS; CRGS, country RGS; URGS, urban RGS.

### 3.3. RGS Service Evaluation under Multiple Travel Modes

Based on the residents' travel behavior (transportation choices), this study conducted a comprehensive evaluation of the RGS services in the GBA. Overall, there is spatial heterogeneity in the RGS services distribution (Figure 9). Overall, there is spatial heterogeneity in the RGS services distribution (Figure 9). The TRGS service hotspots are mainly concentrated along the Pearl River Estuary (Figure 9a), while the CRGS service hotspots are relatively scattered (Figure 9b), and the URGS service hotspots are mainly concentrated at the junction of Guangzhou and Foshan (Figure 9c). This is due to the small radiation range of the TRGS service, and its service level is generally consistent with the population distribution (Figure 3). In contrast, the URGS service radiation range is wide, and its service level is mainly affected by the accessibility of the road network.

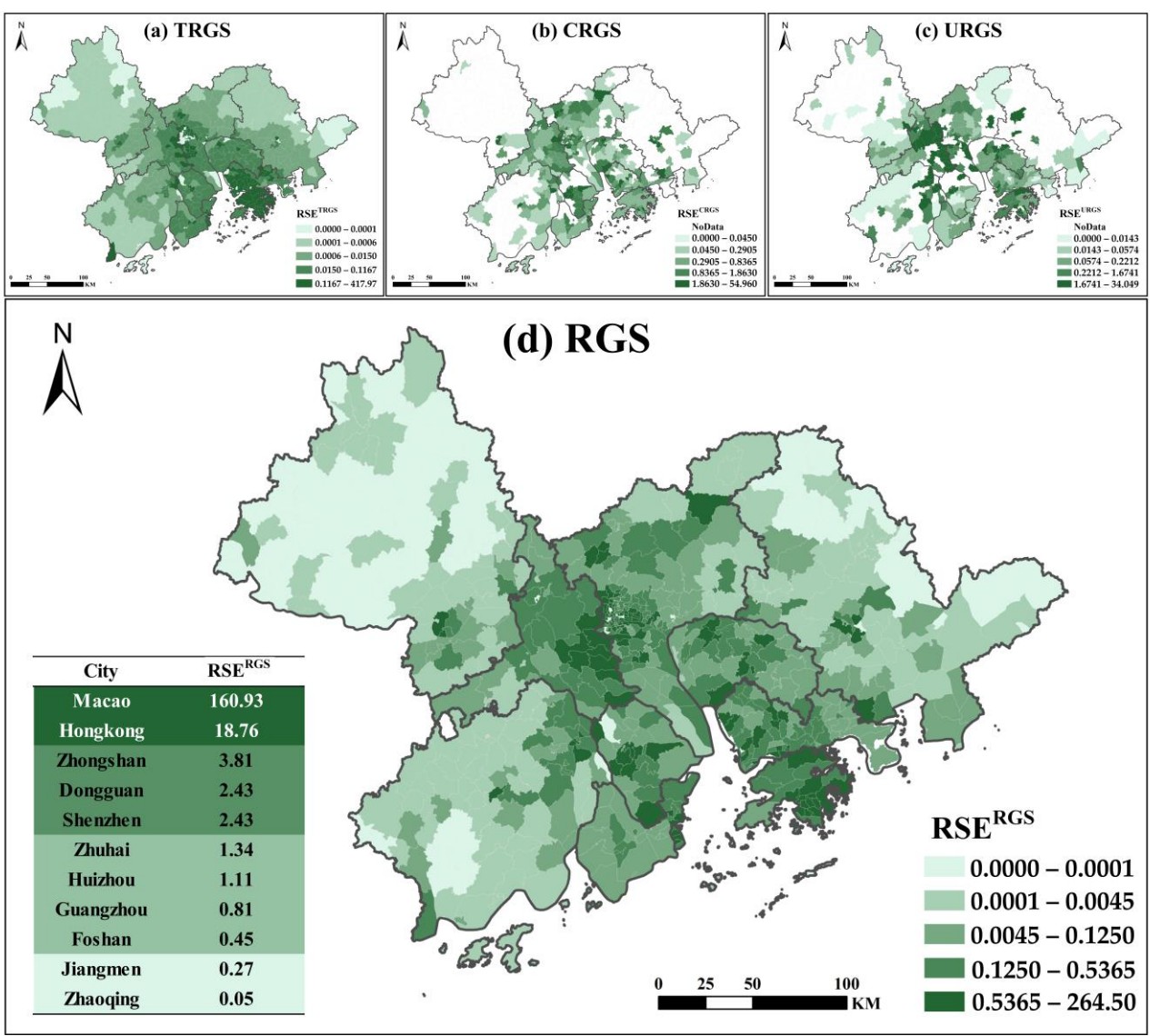

**Figure 9.** RGS service index distribution in the GBA.

Combined with the residents' recreational selection (RGS selection probability), Hong Kong ($RSE^{RGS} = 18.76$) and Macau ($RSE^{RGS} = 160.93$) far outperformed the other nine cities in terms of their RGS service level (Figure 9d). Because these two cities have a long history of modern urbanization, their green infrastructure is well equipped, and their populations are concentrated. Zhaoqing ($RSE^{RGS} = 0.05$) and Jiangmen ($RSE^{RGS} = 0.27$) are at the bottom of the ranking for the RGS services. These two cities are located in the

western part of the GBA where the terrain is mostly mountainous. Figure 7 shows that their three categories of RGS have high per capita ownership, indicating that traffic is the main reason for the low level of the RGS services in these two cities.

## 4. Discussion

An evaluation from the perspective of the residents' travel modes can optimize the planning and maintenance of green space in urban areas. Previous research on the accessibility of UGSs has not classified the green space. In this study, green spaces with recreational value were extracted and classified according to the needs of the residents in different travel modes. Combined with the TIC of multiple travel modes, the service capacity of the RGS was evaluated. This study provides a reference for the refined planning and management of green spaces in mega-urban agglomeration. The findings complement the deficiencies of urban planning policies in the following two aspects: (1) The "Evaluation standard for urban landscaping and greening (GB/T 50563-2010)" classifies the green space coverage into four grades using the greening rate and the vegetation type [38], ignoring the service capacities of different types of green spaces, which cannot truly reflect the recreational service of the green space. The RGS service coverage analysis based on the residents' travel modes can evaluate the recreational service of the green spaces more realistically, and the results help urban planners to allocate urban green space more rationally. (2) The "Standard for Maintaining Quality of Greening Space in City and Town (DB44/T269-2005)" guides green space maintenance by referring to plant growth, climate, and diseases [39], which do not consider the green space recreational services in different regions. Combining the residents' travel modes to evaluate the green space service could determine the pressure of the green space service in each area. The evaluation results can guide the greening manager to maintain the green space, accurately.

In general, the allocation of TRGSs in the GBA is negatively correlated to the urbanization development level (Figure 8) due to the negative impact of urban development on the vegetation [38]. The total coverage of the green space in the GBA accounts for 54.34%, which exceeds that of the TBA (Tokyo Bay Area) (31.64%) and is similar to the NYBA (New York Bay Area) (54.54%), but it still lags behind the SFBA (San Francisco Bay Area) (64.07%) [2,3,40]. It is worth noting that not all green spaces are suitable for recreational activities, so it is more valuable to count green spaces that are defined by recreational functions. The total coverage of the recreational green spaces (RGS) in the GBA is 53.46%, and the per capita ownership is 373.57 $m^2$/p. When different travel patterns and different choices of RGSs are considered, both the TRGSs and the URGSs can cover more than 50% population of the GBA, while CRGSs have a relatively low coverage. Although TRGSs are less developed and smaller in size, their value cannot be ignored. The planners of the NYBA and the SFBA have considered the types of recreation for different green spaces in urban planning, in addition to traditional park green spaces, and they have added small green spaces for short breaks, effectively enhancing the residents' green enjoyment. Urban green space planning has also shown a trend of diversification. The NYBA has a wide range of small parks, including community and pocket parks, to provide citizens with a variety of recreational services and supplement the greening vacancies in built-up areas [3]. However, the planners of the SFBA have transformed street parking spaces into mini green parks, complementing urban parks and effectively enhancing the residents' green enjoyment [41]. Although the planners of the TBA has neglected the construction of small green spaces in the process of urbanization, the local government has planned to develop the recreational function of agricultural land and open private courtyards to make up for this shortcoming [2].

Previous studies on green space services have mostly used buffer zones to analyze the spatial relationship between the residents and the green spaces. For example, Huang et al., used buffer zones to analyze the relationship between urban green space accessibility and urban space expansion [16], while Ingrid et al., used this method to measure the impact of the residents' exposure to green space on health [25]. On the one hand, when it is

compared with a buffer analysis, the application of the TIC in this study considers the traffic factor in the RGS service analysis, which can more objectively reflect the service situation of the RGS. On the other hand, the combination of the RGS classification and the travel mode can realistically evaluate the green space services and provide a reference for the refined planning of green spaces. For example, TRGSs and CRGSs in the central area of the GBA have a strong service capacity (covering more residents), so it is necessary to increase the number of green spaces in the area or moderately limit the number of visiting residents. There are a large number of undeveloped green spaces in the marginal mountainous areas (their service capacity is low). The development of green spaces in this area and the construction of road networks can attract residents to play, so to relieve the pressure of green space service in the central area of the GBA.

This paper has some limitations. First, the RGS service is analyzed using spatial data, and so it ignores the differences in other attributes between different RGSs; for example, the unique landscape, surrounding facilities, etc. The method to integrate more non-spatial factors, such as scenery, culture, and other factors that attract visitors, into the traditional RGS service evaluation can be the focus of subsequent studies.

## 5. Conclusions

RGS service evaluation from the perspective of multiple travel modes can provide planners with guidance for green space conservation, urban planning, and improving residents' green welfare. This study innovatively integrated the residents' travel behavior into the research framework of the UGS, extracted and classified the green spaces with recreational value (RGS), and analyzed RGS services through the TIC under different travel modes. Combined with the residents' travel behavior, a comprehensive evaluation model of RGS service status was then constructed. According to the results, there are sufficient RGS resources in the GBA (i.e., the total RGS area is large). In terms of distribution, the TRGSs are mainly distributed in the northeast, northwest, and southwest edges; the URGSs are mainly concentrated in the mid-western and southern regions; the CRGSs are mostly concentrated in the center of the GBA. Combined with the TIC, the RGS service coverage is higher in the developed cities in (Zhongshan, Shenzhen, Hong Kong, and Macau), but it is weaker in less developed cities (Zhaoqing, Jiangmen), which has a strong correlation with the urban road network and resident distribution density. Combined with the residents' travel behavior (transportation choices) and the recreational selection (RGS selection probability) of multiple travel modes, the comprehensive service level of the RGS shows a gradually decreasing distribution that is centered on the Pearl River Estuary. The RGS service levels of Hong Kong and Macau far exceed that of the other nine cities, and their RGSs can efficiently provide services for the residents. Strengthening conservation should be the focus of their future RGS management. Jiangmen and Zhaoqing have the lowest RGS service levels, but they have a large number of RGSs. The construction of road networks can improve the RGS service level and relieve the pressure on the RGS services in other cities in the GBA. The findings of this study complement the deficiencies of green space-related policies and provide help for the refined management and evaluation of RGSs.

**Author Contributions:** Formal analysis, writing—original draft, C.W.; data curation, J.W.; Writing—review and editing, C.L. (Chunming Li); conceptualization, R.D.; methodology, C.L. (Chencan Lv); data curation, Y.J.; data curation, Y.Z. All authors have read and agreed to the published version of the manuscript.

**Funding:** This research was funded by the Strategic Priority Research Program of the Chinese Academy of Sciences (Program No. XDA23030402), the National Natural Science Foundation of China (Program No. 42277472) and the State Key Laboratory of Urban and Regional Ecology Open Fund (Program No. SKLURE2022-2-5).

**Institutional Review Board Statement:** Not applicable.

**Informed Consent Statement:** Not applicable.

**Data Availability Statement:** Data sharing not applicable.

**Acknowledgments:** This research was supported by funding from the Strategic Priority Research Program of the Chinese Academy of Sciences, the National Natural Science Foundation of China, and the State Key Laboratory of Urban and Regional Ecology Open Fund.

**Conflicts of Interest:** The authors declare no conflict of interest.

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
