# Peer review of "Recreational Green Space Service in the Guangdong–Hong Kong–Macau Greater Bay Area: A Multiple Travel Modes Perspective"

_land, doi:10.3390/land11112072_

Round 1

Reviewer 1 Report

The title is promising, yet the content of the paper needs reconsidering and re-writing.

The paper presents the meticulous study of one area in China, based on data obtained from numerous data-bases, supplemented by mathematical models.

However, the study eventually lacks international perspective which was signaled in the ‘Introduction’ -’...we compared the results with three other global bay areas (Tokyo, San Francisco and New York)’.

The paper lacks also clearly formulated aim and there is no discussion on the authors’ results against previously published studies, especially international and results interpretation is merged into ‘Results’ themselves.

It has not been highlighted to whom the obtained results may be useful.

I attach below some suggestions which may help to improve the paper.

lines 90-105 - contain mixed methodology and choice of subject justification

figure 1 is not self-explanatory

lines 181-191 and table 1 - the definition of urban-type, country-type and township-type recreational green space is unclear

equation 2 is repeated in the place of equation 3

lines 325-327 – it is part of a manuscript template

lines 354-356 – it is results interpretation, it fits better into ‘Discussion’

lines 356-385 – these are not the authors’ results, this section fits better into ‘Discussion’

in lines 405-406 and 413-414 the authors address ‘average’ which has not been reported previously; is it the average calculated for the whole GBA ? If so, what values does the last line in table 4 and 5 present?

lines 452-461 and lines 478-489 - it is results interpretation, it fits better into ‘Discussion’

Reviewer 2 Report

I would like to point out that paper Recreational green space service in the Guangdong–Hong Kong–Macau Greater Bay Area: A multiple travel modes perspective is based on comprehensive research performed and, thus, the information presented are of the importance.

The structure of the paper is, to some point, well organized. The paper is built on appropriate concept, and as authors state “innovatively integrated life circles into research framework of UGS….”. However, this statement is not adequately supported/not clear within manuscript as it is missing wider context/support. Further, I would like to note that one of the limitations of the paper is low quality of communication/presentation. The paper is lacking adequate presentation/writing style. Further, related to referencing, it should be underline that is not in accordance with the Journal guidelines/criteria. Taking into account above mentioned limitations, I have to express my opinion that they affect (adversely) scientific soundness of the manuscript.

I would like to stress that I am not commenting the methods used as it not my key expertise. However, I would like to note that the research is correctly designed. The figures and tables are informative and of importance for understanding the message of the paper.

The paper needs to be improved in order to support publication. Suggestions for author/s:

-          Abstract: What do you consider under „essential ecological infrastructure ...intuitive place...“, „ ..we comprehensively evaluate.” not clear; consider to revise;

-          Introduction: It is stated: “Different types of RGS are adapted to various sightseeing conditions and cannot substitute for each other. For example, small RGSs such as pocket parks are good places to nap, while forest parks are suitable for holiday trips.. (Lines 53,54)- how it comes for “nap”?; please explain/revise;

“At present, providing residents with a variety of leisure options has become a trend in UGS planning....”(line 57) - how? It is suggested to the authors to refer/support this statement;

-          Referencing: generally, main comment is that you refer to several articles within one statement and thus it is difficult to follow, or reference is missing; also, within reference list are not provided required information; please consider revising the way of referencing according to the Journal guidelines (throughout manuscript and the list of reference);

-          Please stay consistent within paper: what do you present: research results or the study performed?

-          Table 1.: you provided green space types; it is not clear how do you come/reference to these types; consider revising it/provide clear reference;

-          Line 324: This statement is part of paper?

-          Line 513: “Our study supplements the current policy..”; “supplements”-what do you consider under this expression? not clear implications, in particular for practice; If you are refereeing/stating policy and strategy, could you please comment;

-          It is suggested to the authors not to use personal pronouns within manuscript - ”we”; consider revising it.

To the authors is suggested to revise paper carefully. In order to support consistence of the paper within the section of Abstract, Introduction and Conclusions should be more clearly emphasized significance of the paper and provide/support to the innovatively stated all together with the value of for similar research in related field.

Please refer to the Authors guidelines while revising the manuscript.

Round 2

Reviewer 1 Report

The authors responded to my suggestions. The aim, the results and the discussion have been improved. However, it still has not been highlighted to whom the obtained results may be useful.

Also the Figure 1 is still not self-explanatory; it means the abbreviations e. g. URGS are explained only in the main text and few pages further but the figure itself lacks ‘note:’.

Reviewer 2 Report

I would like to point out that the authors of the paper Recreational green space service in the Guangdong–Hong Kong–Macau Greater Bay Area: A multiple travel modes perspective corrected and improved the manuscript. I would like to stress that I am not commenting the methods used as it not my key expertise. However, I would like to note that the research is correctly designed. The figures and tables are informative and of importance for understanding the message of the paper.

In spite of this, I have to underline that I assigned major revision due to the fact that some of the recommendations and suggestions have not been considered appropriately. It is difficult to follow the paper as the paper is missing consistence related firstly to the quality of presentation. Also, I would like to note that I am of the opinion that paper needs proofreading.

Suggestions for authors:

Structure of the paper: Consider revising as some parts are only with one/two sentences (Section 2); Lines 228,229- Section 2.2. Data and methods; Sentences: “Fig. 2 presents the RGS service evaluation workflow of this study. Each step and method would be explained in detail in this section.” – consider revising, you could probably give a statement (researcher attitude) related to the workflow.

It was suggested to the authors not to use personal pronouns within manuscript -” we”; consider revising it; the authors responded, “we changed some of the "we" and made sure the sentences were smooth.”; to the authors is suggested to correct the paper and not to use personal pronouns “we”, “our”.

Line 839 (Conclusions): “Our study supplements the current policy.”; “supplements” - what do you consider under these expressions? not clear implications, in particular for practice; If you are refereeing/stating policy and strategy, could you please comment; Note: the authors provided response to the comment but still within Conclusions stayed expression” supplements”; consider revising it.

Referencing: generally, it is improved but still in some parts needs revision regarding the comment that authors refer to several articles within one statement and thus it is difficult to follow or reference is missing; for example, line 839: “Our study supplements the current policy of urban green spaces [31,45,46] and provides guidance for improving residents’ RGS recreational experience.” -not clear, why you are referring to the references 31, 45 and 46?  

To the authors is suggested to revise paper carefully in order to support consistence of the paper related firstly to the quality of presentation. It is difficult to follow the paper and thus all sections should be more clearly presented, including appropriate referencing. Further, within the sections of Abstract, Introduction and Conclusions should be more clearly emphasized significance of this paper. Taking into account above mentioned, this will contribute to the improvement of scientific soundness of the manuscript and meet the criteria for publishing.  
